# Motion Compensation for Long Integration Times and DoA Processing in Passive Radars

Anabel Almodóvar-Hernández, David Mata-Moya, María-Pilar Jarabo-Amores *, Nerea Rey-Maestre and María Benito-Ortiz

Signal Theory and Communications Department, University of Alcalá, 28805 Alcalá de Henares, Spain
* Correspondence: mpilar.jarabo@uah.es

**Abstract:** In this work, a multistage target motion compensation solution for long integration times and the direction of arrival processing in geostationary-satellite-based passive radars is presented. Long integration processing intervals are considered to compensate for the associated propagation loss, but during this time target dynamics can extend the backscattering in more than one range or Doppler cell. To control the gain-processing reduction, a combination of detection, tracking, feature extraction, and filtering techniques is designed to provide automatic adaptation to each unknown target dynamic in the area of interest. The proposed methodology is validated with real data acquired by the passive radar demonstrator developed by the University of Alcalá (IDEPAR), and the results confirm that target monitoring exploiting digital video broadcasting-satellite (DVB-S) signals is clearly improved.

**Keywords:** passive radar; DVB-S; target motion compensation

## 1. Introduction

Passive radars (PRs) encompass all those techniques used for the detection and estimation of target parameters making use of Illuminators of Opportunity (IoO) instead of a dedicated transmitter [1]. PR are not subject to electromagnetic emissions legislation and do not cause an environmental impact or electromagnetic compatibility problems. These systems involve reduced cost, portability potential, a lack of electromagnetic spectrum allocation, and robustness against interception and jamming. However, the absence of the dedicated transmitter is also the origin of the challenges to be solved due to the use of signals not designed for detection purposes, and of uncontrolled transmitters whose locations, transmitted powers, and coverages have been designed to meet the quality of service requirements imposed by the radio services they belong to.

Broadcasting systems such as FM [2], digital audio broadcasting (DAB) [3], digital video broascasting terrestrial and satellite (DVB-T and DVB-S, respectively) [4–7] are IoOs of great interest that are under study due to their different, and, in many cases, complementary characteristics, related to IoO availability, radiated power, bandwidth, and their impact in PR coverages and resolutions. PRs have proven their capabilities in air, land, and maritime traffic monitoring and imaging (inverse synthetic-aperture radar, ISAR) [8–13].

Over the last few years, the exploitation of satellite IoOs in PRs has been under intense investigation. One of its main advantages is based on the large number and diversity of available constellations. Different waveforms, frequencies, bandwidths, and radar geometries can be exploited for improving PR performance. Satellite systems provide high availability, global coverage, and near total invulnerability to natural disasters or physical attacks. However, these types of systems also have a number of drawbacks such as reduced signal power at ground level due to the large distance to the transmitter, and the requirement of high gain antennas, which can compromise the detection capabilities due to the inverse relationship between antenna gain and beamwidth. As a result, a

significant reduction in PR coverage is expected by employing satellite IoOs compared to using ground-transmitted signals such as FM, DAB, or DVB-T [14]. In spite of these drawbacks, the exploitation of DVB-S signals' potential benefits is relevant , and many studies can be found in the radar literature.

DVB-S satellites are located in a geosynchronous equatorial orbit (GEO) providing an almost static radar geometry. As a result, radar signal processing is simplified, thus avoiding the need for transmitter motion compensation. The transmitted waveform corresponds to a B1-type signal, whose ambiguity function (AF) is a thumbtack that spreads throughout the delay-Doppler plane but at range bins far away from the expected detection coverages . DVB-S based on PR systems have been considered for the detection of maritime targets [15], drones [16–18], ground vehicles [7,19–21], inverse synthetic aperture radar (ISAR) imaging [22], and forward scatter for aerial target detection [23].

The theoretical integration gain associated with coherent processing is calculated as the product of the integration time $T_{int}$ and the signal bandwidth (*BW*). Wide *BWs* can give rise to Doppler spread effects in targets' contributions because of the different Doppler shifts associated with different carrier frequencies; long $T_{int}$ values can generate target Doppler and range migration [24]. These effects produce integration gain losses that limit the detection and localization performances.

Focus-before-detection (FBD) methods are able to perform long-time coherent integration for weak targets without prior knowledge of the target motion parameters. One of the best well-known methods to solve this type of problem is the Radon Fourier transform (RFT), which achieves high integration times of the target whose range migration is significant, by parametric signal modelling and projecting echoes into a low-dimension parameter space. On the other hand, there is also the generalized Radon–Fourier transform method (GRFT), which provides a general method for high coherent integration times [25,26]. However, these methods do not solve the Doppler migration problem. To solve this problem, the modified Radon Fourier transform (MRFT) for long-time coherent integration is presented in [27,28], which searches the Doppler rate during the integration of pulses. The MRFT integrates the target echo energy by searching for the range, Doppler, and Doppler rate of the target. This algorithm is able to correct both range and Doppler migration problems and significantly improve the signal-to-noise ratio (SNR) of the signal echo.

In [29] classical compensation methods are compared such as the use of Keystone transform (KT), which is used in a multitude of applications such as SAR (synthetic aperture radar), or the envelope alignment based on envelope correlation for range compensation and the Chirp–Fourier transform (CFT) in Doppler compensation. The CFT is also proposed in [16,27]. According to the paper [30], for high integration times, it is proposed to make use of long-time integration Fourier transform (LIFT) to obtain high coherent processing gain. After clutter cancellation, the LIFT is proposed to achieve azimuth compensation. In addition, the Doppler shift is defined as the relative acceleration, which must be calculated before applying for Doppler compensation. If the estimation of this parameter is not precise, a residual white dispersion will be obtained.

The presented solution in this work allows for an improvement in detection and tracking capabilities for non-collaborative targets with unknown dynamic implementing target motion compensation in both Doppler and bistatic range domains. The signal processing scheme is based on a multistage range-Doppler migration compensation architecture that combines detection, tracking, and feature-extraction techniques to provide automatic adaptation to each target dynamic in the bistatic scenario of interest. The main steps are as follows:

1. First detection and tracking performances are used to characterized the targets movements during the acquisition time to define a set of discrete possible target bistatic accelerations.
2. Doppler migration compensation is performed using a filter bank generating a set of Range-Doppler Maps (RDMs) for each single element of the surveillance array and each considered target bistatic acceleration. As the integration gain is expected to be

increased, the detector and tracker are again performed. The bistatic tracker results for each acceleration are combined to obtain an improved target trajectory declaration.

3.  For each target trajectory, the acceleration and velocity information are extracted to perform the Doppler and range compensation at a single radiation element level.
4.  The final signal-to-interference ratio (SIR) values will outperform those achieved without target motion compensation techniques, allowing for detection capabilities improvement and range and azimuth-tracking-accuracy enhancement.

The proposed radar signal processing scheme has been validated using data acquired by the PR demonstrator IDEPAR (Improved DEtection techniques for PAssive Radars), developed in the University of Alcalá for DVB-T and DVB-S exploitation [31–33], in a bistatic semiurban scenario with a collaborative terrestrial target. This controlled scenario allows one to perform a detailed study of the detection improvement in the different processing stages. The results confirm the PR performance improvement even when long-time integration time is considered.

The rest of the paper is structured as follows: in Section 2, a brief description of the PR operating principle is carried out; in Section 3, the main parameters of a bistatic radar scenario are detailed; in Section 4, the problem of used long integration intervals is addressed; in Section 5, the IDEPAR demonstrator is described; in Section 6, the proposed solution is explained; in Section 7, the experimental results for validation purposes are presented; and, finally, in Section 8, the conclusions are summarized.

## 2. Passive Radar Operating Principle

The basic geometry of a PR based on satellite IoO is depicted in Figure 1: $L$ is the baseline or IoO-PR distance; $R_T$ is the IoO-target distance; $R_R$ is the PR-target distance; $\beta$ is the bistatic angle; and $\vec{V}_{sat}$ and $\vec{V}_{tg}$ are the velocity vectors of the satellite and the target, respectively. Since DVB-S satellites have a GEO orbit, they can be considered stationary, which simplifies PR scenario geometry and guarantees the availability of stable illuminating signals.

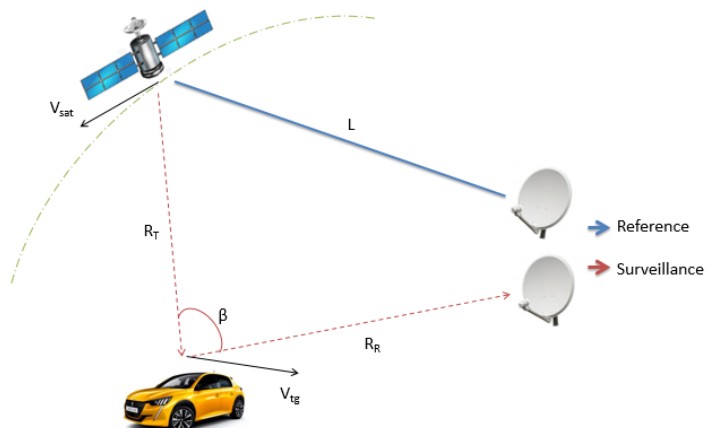

**Figure 1.** Basic geometry of a satellite IoO-based passive radar.

Two receiving chanels are used: the reference one, which acquires the direct signal from the IoO; and the surveillance one, which captures the echoes from the targets when they are illuminated by the IoO. If the bistatic range origin is located at the PR receiver, the target echo delay is expressed as $t_{target} = t_T + t_R - t_L$. $t_{target}$ is calculated from the bistatic range, $R_{bis} = R_T + R_R$, and the baseline as $t_{target} = (R_{bis} - L)/c$, being $c$ the light speed; associated propagation delays are $t_L = L/c$, $t_T = R_T/c$ and $t_R = R_R/c$.

The cross ambiguity function (CAF) is the main coherent processing stage. It performs the matched filter, allows for the estimation of the bistatic range and Doppler shift of the target, and provides an integration gain whose theoretical value equals the product

$T_{int} \cdot BW$. The acquisition time, $T_{acq}$, is divided into coherent processing intervals (CPIs) of duration $T_{int}$ for performing signal processing (Figure 2). The time between CPIs is the pulse repetition frequency (PRI), which can be lower than $T_{int}$ to process overlapped CPIs, equal to $T_{int}$ for processing all of the available data, or higher than $T_{int}$ to obtain a compromise solution between the computational cost and the refresh rate.

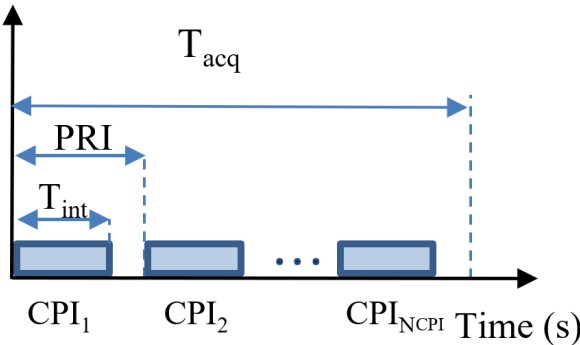

**Figure 2.** Time parameters for PR data processing.

To generate the CAF, delayed and Doppler-shifted copies of the reference signal are correlated with the surveillance one (1). This operation is usually performed at base-band after analog-to-digital conversion. At this point, the reference and surveillance sequences are $s_{ref}[n]$ and $s_{surv}[n]$ (1) [2,34].

$$S^{CAF}[m, p] = \sum_{n=0}^{N-1} S_{ref}^*[n - m] \cdot S_{surv}[n] \cdot \exp\left(-j2\pi\frac{p}{N}n\right) \tag{1}$$

where

- $N = T_{int} \cdot f_s$, is the number of sample acquired during $T_{int}$ (s), which defines the duration of the coherent processing interval (CPI), and $f_s$ (Hz) is the sampling frequency;
- $m = 0, \dots, M - 1$ is the time bin associated with a delay $t_m = m/f_s$, with $M$ being the number of range cells for a given instrumented radar coverage and $t_m \in [0, T_{int}]$.
- $p = -P/2, \dots, (P/2) - 1$ is the Doppler-shift, corresponding to $f_D = f_s(p/N)$, with $P$ being the number of Doppler shift cells for a the Doppler interval of interest defined in $[f_s(-(P/2)/N), f_s(((P/2) - 1)/N)]$.

For each target, the result of the CAF is the ambiguity function (AF) of the transmitted signal, scaled and shifted to be centered on the target bistatic time delay and Doppler shift. The stationary targets' main contributions appear along the zero Doppler line of the CAF output domain (the Range-Doppler Map, RDM), although, depending on the signal AF, their contributions can spread throughout all of the RDM. This is the case of the DVB signals (terrestrial and satellite), which are B1-type signals.

So far, defined signal models have a focus on the target. Actually, the reference signal is noisy, and it can be affected by multipaths. The surveillance signal also includes the direct signal from the IoO (direct path interference, DPI) and clutter. These elements are filtered by zero Doppler rejection techniques, so the detection in the zero Doppler line is not considered by focusing this work on moving targets' returns and the reduction of integration gain losses when long integration times are used.

## 3. Bistatic Scenarios with Stationary PR and IoO

In the study case, the PR receiver and the IoO are stationary. The target bistatic Doppler shift depends on the projection of the target speed vector $\vec{v}_T = v_T \hat{v}_T$ on the bistatic plane, $\vec{v}_{tg} = v_{tg}\hat{v}_{tg}$, and it is calculated as $f_D(Hz) = (2 \cdot v_{tg}/\lambda) \cdot \cos(\psi_{tg}) \cdot \cos(\beta_{tg}/2)$, with $\psi_{tg}$ being defined by the projected target speed vector, $\hat{v}_{tg}$; the bistatic bisector, $\beta_{tg}$; and the

target bistatic angle [1] (Figure 3). In this case, the target bistatic delay depends on time $t_{tg}(t) = t_T(t) + t_R(t) - t_L = (R_{bis}(t)/c) - t_L$.

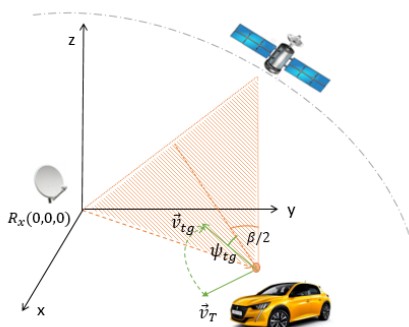

**Figure 3.** Projections on the passive radar, illuminator of opportunity, and target plane.

$R_{bis}(t)$, according to the Weierstrass approximation principle, can be expanded to the second order (Equation (2)) defining $R_{bis0}$ as the bistatic range at the reference time, $v_{bis0} = f_{D0}\lambda$ as the bistatic velocity and Doppler shift at the reference time $t_0$, and $a_{bis} = f_{Dr}\lambda$ as the bistatic acceleration and Doppler shift rate [28]. For the CPI processing, a relative time $\tau = t - t_0$ is used assuming $t_0 = n_{CPI}PRI$ for redefining $R_{bis}$ as (3).

$$R_{bis}(t) = R_{bis0} - f_{D0}\lambda(t - t_0) - \frac{f_{Dr}\lambda(t - t_0)^2}{2} \tag{2}$$

$$R_{bis}(\tau) = R_{bis0} - f_{D0}\lambda\tau - \frac{f_{Dr}\lambda\tau^2}{2} \tag{3}$$

For a IoO passband signal $S(\tau) = Re\{\tilde{s}(\tau + t_L) \exp(j2\pi f_c(\tau + t_L))\}$, the reference and surveillance baseband signals are, respectively, expressed in (4) and (5) [35], where:

- $f_c$ is the carrier frequency.
- $\zeta_{ref}$ and $\zeta_{surv}$ are the complex gain factors for the reference and the surveillance signals, respectively. The propagation effects and clutter and target contributions are considered, but possible reference channel multipaths are discarded.
- $\tau_{tg}(\tau) = (R_{bis}(\tau)/c) - t_L$.
- $\tau_{tg0} = (R_{bis0}/c) - (L/c)$ is the target bistatic delay at $\tau = 0$ or $t = t_0$.
- $\phi_{t0} = -j2\pi f_c \tau_{tg0}$ is the phase constant associated with $\tau = 0$ or $t = t_0$.

$$\tilde{s}_{ref}(\tau) = \zeta_{ref} \cdot \tilde{s}(\tau) \tag{4}$$

$$
\begin{aligned}
\tilde{s}_{surv}(\tau) &= \zeta_{surv} \cdot \tilde{s}(\tau - \tau_{tg}(\tau)) \cdot \exp(-j2\pi f_c \tau_{tg}(\tau)) \\
&= \zeta_{surv} \cdot \tilde{s}(\tau - \tau_{tg}(\tau)) \exp\left(j2\pi f_c\left(\frac{f_{D0}}{f_c}\tau + \frac{f_{Dr}}{2f_c}\tau^2\right) + \phi_{t0}\right) \\
&= \zeta_{surv} \cdot \tilde{s}(t - \tau_{tg}(\tau)) \exp\left(j2\pi\left(f_{D0}\tau + \frac{f_{Dr}\tau^2}{2}\right) + \phi_{t0}\right)
\end{aligned}
\tag{5}
$$

## 4. Long Integration Time Passive Radar Processing

In a classical PR, the processing scheme is limited by the time a target remains in the same resolution cell. In this case, target bistatic time delay and Doppler shift can be assumed to be constant during the CPI, and the CAF output, expressed in (6), can be obtained from Equation (1) being $n = (t - t_0)f_s$, $t_0 = n_{CPI} \cdot PRI$, $m_{tg0} = (\tau_{tg0}f_s)$, $p_{tg0} = (f_{D0}N/f_s)$ and $E_T$ the energy of $\tilde{s}(t)$.

$$S^{CAF}\left[m_{tg0}, p_{tg0}\right] = \sum_{n=0}^{N-1} S_{ref}^*\left[n - m_{tg0}\right] \cdot S_{surv}[n] \cdot \exp\left(-j2\pi \frac{p_{tg0}}{N} n\right)$$
$$= \xi_{ref}^* \cdot \xi_{surv} \cdot E_T \cdot \exp\left(\phi_{t0}\right) \tag{6}$$

Range walk (the movement of the target during $T_{int}$ exceeds the range resolution) and Doppler walk (the acceleration of the target produces a Doppler cell migration during $T_{int}$) give rise to target contributions' spreading along range and Doppler and integration gain losses. The associated reduction of signal to interference level will deteriorate the probability of detection ($P_D$) for the selected probability of false alarm ($P_{FA}$). Additionally, the spreading effects complicate the estimation of the detected targets' bistatic range and Doppler and reduce the system's accuracy.

To avoid range walk, the target movement during $T_{int}$ should be smaller than the bistatic range resolution cell, $\Delta_{R_b} = c/BW$. The main parameters are the radial components of the target velocity vector projected on to the bistatic plane, $\vec{v}_{tg} = v_{tg}\hat{v}_{tg}$, with respect to the IoO, $v_{tg,rT}$, and the PR receiver, $v_{tg,rR}$. These components depend on the target trajectory described by the unitary vector $\hat{v}_{tg}$. Taking into consideration that $v_{tg,rT}$ and $v_{tg,rR}$ are lower than or equal to $v_{tg}$, a worst case $T_{int}$ maximum value can be estimated as $T_{int,max} = c/(2BWv_{tg})$. This expression assumes a constant $v_{tg}$ during $T_{int}$.

The bistatic velocity resolution depends on $f_c$ and $T_{int}$: $\Delta_{v_{bis}} = c/(f_c T_{int})$. If target bistatic acceleration is assumed to be constant during the integration time, the maximum value that ensures the target remains in the same Doppler cell can be obtained as $a_{bis,max} = c/(2f_c T_{int}^2)$. The dependence on the signal frequency is an important factor, especially for DVB-S signals, due to the high carrier frequencies (in the Ku band).

If $T_{int}$ is higher than $T_{int,max}$ and the target bistatic acceleration is higher than $a_{bis,max}$, $S^{CAF}\left[m_{tg0}, p_{tg0}\right]$ has to be modified, as seen in Equation (7), where $m_{tg} = (\tau_{tg}f_s)$, to include the spreading effects.

$$S^{CAF}\left[m_{tg0}, p_{tg0}\right] = \sum_{n=0}^{N-1} S_{ref}^*\left[n - m_{tg0}\right] \cdot S_{surv}[n] \cdot \exp\left(-j2\pi \frac{p_{tg0}}{N} n\right)$$
$$= \sum_{n=0}^{N-1} \xi_{ref}^* \tilde{s}^*\left[n - m_{tg0}\right] \cdot \xi_{surv}^* \tilde{s}^*\left[n - m_{tg}\right] \exp\left(j2\pi f_c \frac{m_{tg}}{f_s}\right) \cdot \exp\left(-j2\pi \frac{p_{tg0}}{N} n\right) \tag{7}$$
$$= \xi_{ref}^* \xi_{surv} \exp\left(\phi_{t0}\right) \sum_{n=0}^{N-1} \tilde{s}^*\left[n - m_{tg0}\right] \tilde{s}\left[n - m_{tg0} + \frac{f_{D0}n}{f_c f_s}\right] \exp\left(j\pi \frac{f_{Dr}n^2}{f_s}\right)$$

## 5. IDEPAR Demonstator for DVB-S

IDEPAR is a PR technological demonstrator developed in the University of Alcalá [5,31,36]. The acquisition chain is composed of two universal software radio peripheral (USRP) software defined radio (SDR) devices. Each one can perform full-coherent digitalization of four channels with a maximum instantaneous bandwidth of 100 MHz per channel.

An $N_a = 7$ element surveillance array was built using low-cost commercial single elements. Each single antenna uses a different acquisition channel. The eighth channel was reserved for acquiring the reference signal using a parabolic reflector antenna with higher gain. The surveillance array is composed of axially corrugated conical horns and low noise blocks (LNBs) with conversion gains greater than 50 dB (Figure 4a). Horizontal or vertical polarization can be selected [37]. The simulation results presented in [31] showed that the single element achieves a maximum realized gain of 13.9 dBi with a symmetrical HPBW= 38.4° in both E and H planes (Figure 4b). The planar array architecture is shown in Figure 5. The main parameters are an angular resolution of 3.2°; maximum directivity $D_{max} = 22.3$ dBi; and a side lobe level (SLL) of 12.2 dB in the worst case [31]. For an azimuth pointing direction of 12°, the grating lobe appears at −14.3°, reaching almost the same directivity level as the main desired lobe. The steering margin to unambiguous DoA (Direction of Arrival) estimation must be limited to $[-12°; 12°]$, reducing the azimuth coverage area to 24°.

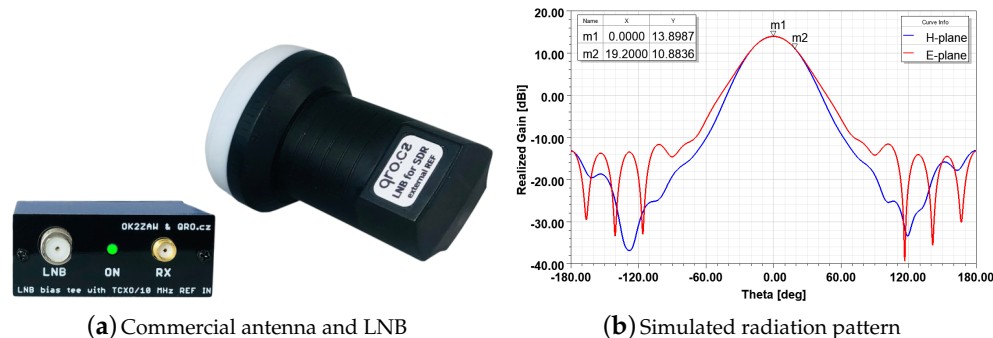

(**a**) Commercial antenna and LNB      (**b**) Simulated radiation pattern

**Figure 4.** Characterization of commercial axially corrugated conical horn [31].

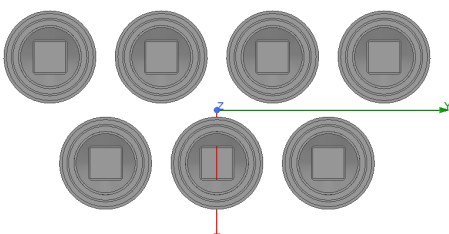

**Figure 5.** Surveillance planar array [31].

IDEPAR signal processing scheme, proposed in [8], is shown in Figure 6:

- The reception stages include RF-front ends and digitalization processes. In the processing stages, zero Doppler interference rejection techniques are applied to obtain the snapshot defined in the time domain: $\mathbf{s}_{surv}[n] = [s_{surv,1}[n], s_{surv,21}[n], \dots, s_{surv,N_a}[n]]^T$, where $T$ denotes the transpose vector. These techniques are applied to reduce DPI (direct path interference), multipath, and stationary clutter sources.

- The acquisition time is divided into CPIs of duration $T_{int}$, and independent CAFs are calculated for each single radiating element signal $s_{surv,i}[n]$, $i = 1, \dots, N_a$. These are denoted as CAFs $\mathbf{s}_{CAF}[m, p] = [s_{CAF,1}[m, p], s_{CAF,2}[m, p], \dots, s_{CAF,N_a}[m, p]]$.

- Digital beamforming is applied in the CAF domain to generate $N_{BF}$ orthogonal beams pointing to azimuths $\Phi = [\phi_{BF,1}, \phi_{BF,2}, \dots, \phi_{BF,N_{BF}}]$ within the azimuth coverage without grating lobes ambiguities. The use of orthogonal beams guarantees that a signal arriving along the maximum radiation axis of a beam will have no output in any other beam, and that a signal that is not along the maximum radiation axis will appear in the side lobes of the other beams [38]. The result is a three dimensional matrix called $\mathbf{S}_{CAF} = [s_{CAF}[m, p, \phi_{BF,1}], \dots, s_{CAF}[m, p, \phi_{BF,N_{BF}}]]$.

- 3D reference window CFAR constant false alarm rate: (a) For each $[m, p]$ pair, the cell under test (CUT) is the cell $s_{CAF}[m, p, \phi_{BF,i}]$ for the $\phi_{BF,i}$ value where the maximum power of the echo was received; (b) a 3D reference window with dimensions $R_R x R_D x N_{\phi_{BF}}$ (range x Doppler x steering angle) was generated using the neighbour cells around the CUT, excluding the guard cells defined along range and Doppler for all of the steering angles. This 3D reference window also excludes $s_{CAF}[m, p, \phi_{BF,i}]$, $i = 1, \dots, N_{\phi_{BF}}$.

- The detection matrix generated by the 3D reference window CFAR is applied to a bistatic tracker. The linear functions that govern the dynamic model in the bistatic range-Doppler domain, and the direct relation with the measurement vector, make the linear Kalman filter the optimum solution in terms of root-mean-square error (RMSE) [39]. Tracking results consist of declared target trajectories that can be used to estimate target parameter dynamics or to provide the DoA estimation stage input.

- The DoA estimation is carried out in a new beam-space with better accuracy and better resolution, using beamforming weights calculated for maximizing the directivity.

The resulting 3D (range-Doppler-azimuth) estimations are processed by a Cartesian tracker to obtain the target's location in the (x-y) plane. The coordinate transformation is performed by non-linear functions that depend on the definition of the PR scenario. Due to the non-linearities of the measurement model, the optimal filter is also non-linear, and suboptimal solutions can be considered: coordinate transformation in combination with a linear Kalman filter [40], an extended Kalman filter [41], or unscented Kalman filter [42]-based solutions. Detection to track the association process for confirmed targets is performed directly from the Cartesian tracks' estimations.

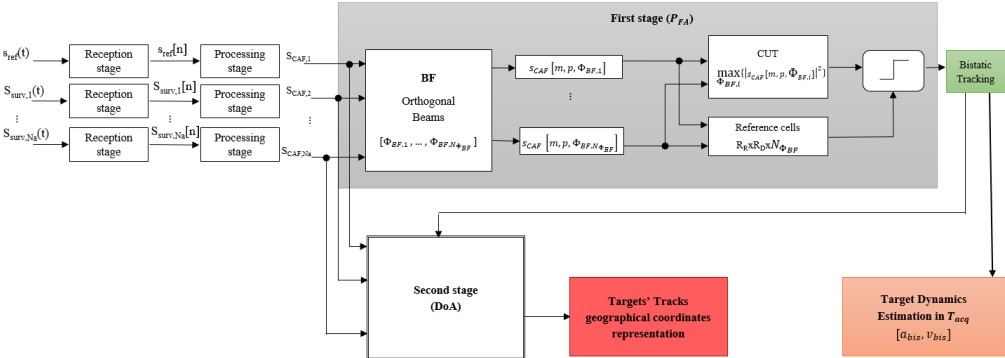

**Figure 6.** IDEPAR array signal processing processing scheme [8].

## 6. Multi-Stage Range-Doppler Migration Compensation

In expression (7), the spreading effects are detailed due to the target bistatic delay $m_{tg0} = (\tau_{tg0} f_s)$ and its variation during the $T_{int}$. Then, the main objective is the estimation of target dynamics (range, Doppler, and Doppler rate parameters) in each CPI to perform coherent integration reducing integration losses. This technique is applied to every detected target, so targets must be detected before starting the compensation process. To implement target motion compensation techniques based on inverse discrete Fourier transform (IDFT) and discrete Fourier transform (DFT), targets dynamic characterized by constant bistatic acceleration and rectilinear trajectory in the bistatic range and Doppler domain during the CPI are assumed. The proposed strategy is based on the following steps:

- **First Stage:** Although a first detection at single element level would simplify the process, bistatic tracker results after beamforming technique are used (Figure 6) due to the high propagation losses and the reduced integration gain. As target contributions are spread throughout the 3D matrix $\mathbf{S}_{CAF}$, the probability of detecting such a target can be improved by increasing the probability of false alarm and using the bistatic tracker to control the excess of false alarms.

- **Doppler migration compensation filter bank:** Once possible target trajectories are declared in the area of interest, the associated bistatic velocities and accelerations can be used for the definition of $N_{fDr}$ discrete $a_{bis}$ values, associated with Doppler rates from $f_{Drmin}$ to $f_{Drmax}$, which will be considered to perform Doppler migration compensation and to improve target detection capabilities. $N_{fDr}$ should be selected as a compromise between accuracy and computational cost.

  For one specific value $f_{Dr}^i$, the compensation technique is based on applying inverse discrete Fourier transform (IDFT) expressed in Equation (8). In other words, IDFT performing across the $p$-axis ($IDFT_p$) or Doppler axis of the RDMs. If the expression

of $S_{surv}[n]$ is substituted, the equality (9) is obtained where a $f_{Dr}$-dependent term responsible for the Doppler walk appears.

$$\text{IDFT}_p(S^{CAF}[m,p])$$

$$= \sum_{p=-P/2}^{(P/2)-1} \sum_{n=0}^{N-1} S_{ref}^*[n-m] \cdot S_{surv}[n] \cdot \exp\left(-j2\pi\frac{p}{N}n\right) \cdot \exp\left(j2\pi\frac{p}{N}n\right) \tag{8}$$

$$= \sum_{n=0}^{N-1} S_{ref}^*[n-m] \cdot S_{surv}[n]$$

$$\text{IDFT}_p(S^{CAF}[m,p])$$

$$= \sum_{n=0}^{N-1} S_{ref}^*[n-m] \cdot \xi_{surv} \cdot \tilde{s}[n-m_{target}[n]] \exp\left(j2\pi\frac{f_{D0}n}{f_s}+\phi_{t0}\right) \exp\left(j\pi\frac{f_{Dr}^i n^2}{f_s}\right) \tag{9}$$

Then, the filter $H_D$, defined in (10), will be implemented to obtain a corrected version of the IDFT of the CAF in (11), resulting in the convolution of the reference signal with the corrected version of the surveillance one $S'_{surv}[n]$.

$$H_D[n, f_{Dr}^i] = \exp\left(-j\pi\frac{f_{Dr}^i n^2}{f_s}\right) \tag{10}$$

$$\text{IDFT}'_p(S^{CAF}[m,p])$$

$$= \sum_{n=0}^{N-1} S_{ref}^*[n-m] \cdot \xi_{surv} \cdot \tilde{s}[n-m_{target}[n]] \cdot \exp\left(j2\pi\frac{f_{D0}n}{f_s}+\phi_{t0}\right) \tag{11}$$

$$= \sum_{n=0}^{N-1} S_{ref}^*[n-m] \cdot S'_{surv}[n]$$

- **Second Stage:** The inputs of each filter $H_D[n, f_{Dr}^i]$ are the CAFs generated by each single antenna of the surveillance array (Figure 7). The detection and tracking stages are performed with higher $P_{FA}$ requirements to the outputs of each filter $H_D[n, f_{Dr}^i]$, where an SIR improvement is expected to be associated with Doppler migration compensation. The $N_{f_{Dr}}$ tracking results are combined using a weighted centroid of the target detected pixels and their SIRs in each CPI to obtain target trajectories in $T_{acq}$ with more accuracy.

  At the output of the considered detector, the detection matrix is generated, and a target can be observed in a set of cells depending on its size and kinematics. The target position (range-Doppler in the RDM) could be estimated by the data extractor. This extractor applies algorithms such as the flood-fill algorithm, which connects all of the hints associated with a target and estimates the target position in the system coordinates. These estimated values define a point in the input 2D space, which is denoted as the plot associated with the target. Once the connected hints are detected, the centroid related to each blob is calculated as its center of mass considering the corresponding pixels intensity or outputs of the envelope detector.

  For each declared target trajectory, a mean value of Doppler Shift rate during $T_{acq}$, $\overline{f_{Dr}}$, is estimated to define a constrained version of the filter bank with $N'_{f_{Dr}} < N_{f_{Dr}}$ filters designed for $f_{Dr}^{c,1}, \ldots, f_{Dr}^{c,N'_{f_{Dr}}}$ values close to $\overline{f_{Dr}} = \overline{a_{bis}}/\lambda$ (Figure 8). For the next stage and for each CPI, only the corrected CAFs associated with the filter designed the $f_{Dr}^{c,i}$ that provides the highest SIR are selected. This strategy is based on refining the Doppler migration compensation in case the target bistatic acceleration is not constant during $T_{acq}$.

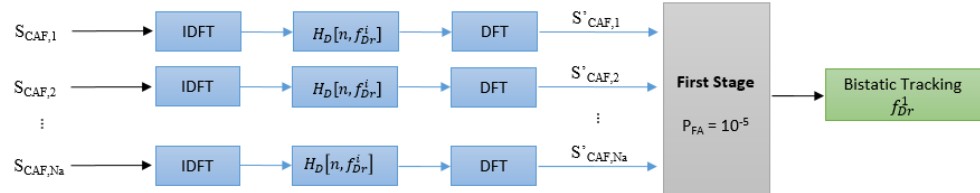

**Figure 7.** Doppler migration compensation scheme for $f_{Dr}^i$.

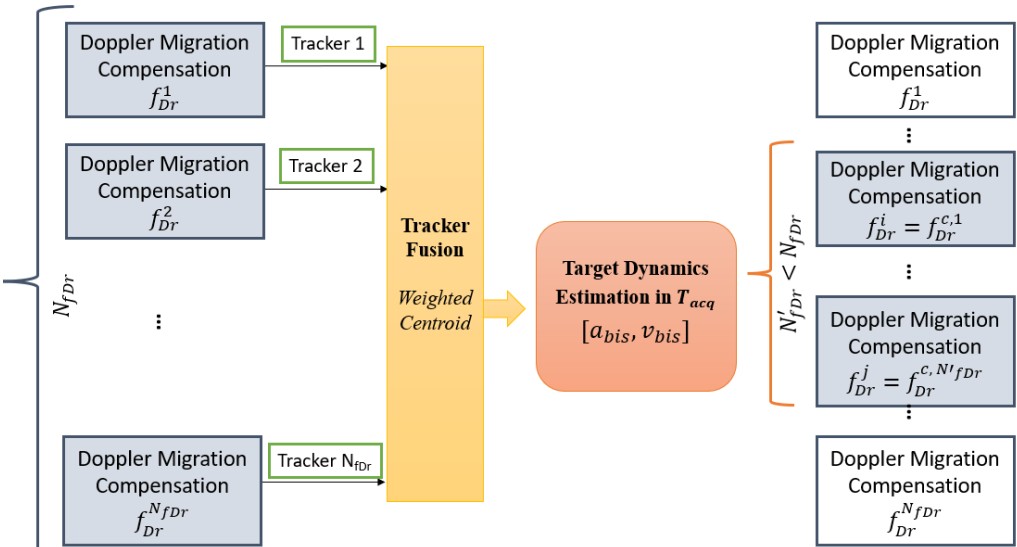

**Figure 8.** Doppler migration filter banks for a specific target.

- **Third Stage:** Before range migration correction is performed, the discrete Fourier transform (DFT) of expression (11) is used and detailed in (12). In this case, DFT is performed across the $m$-axis ($DFT_m$) or bistatic range axis of the RDMs. With a change of variable $y = n - m$, the result (13) can be expressed in the function of the DFT of the surveillance signal corrected version.

$$\text{DFT}_m(\text{IDFT}'_p(S^{CAF}[m, p])') = \sum_{m=0}^{N-1} \sum_{n=0}^{N-1} S^*_{ref}[n - m] \cdot S'_{surv}[n] \exp\left(-j2\pi \frac{k}{N} m\right) \quad (12)$$

$$
\begin{aligned}
\text{DFT}_m(\text{IDFT}'_p(S^{CAF}[m, p])') \\
= \sum_{y=0}^{N-1} S^*_{ref}[y] \exp\left(j2\pi \frac{k}{N} y\right) \sum_{n=0}^{N-1} S'_{surv}[n] \exp\left(-j2\pi \frac{k}{N} n\right) \\
= \text{IDFT}_y(S^*_{ref}[y]) \cdot \text{DFT}_n(S'_{surv}[n])
\end{aligned}
\quad (13)
$$

The use of the expression of $S'_{surv}[n]$ and the properties of the DFT leads to a $f_{D0}$-dependent exponential, responsible for the range migration, defined in (14). Then, the filter $H_R$, (15) is implemented to obtain an updated version of the surveillance signal, $S''_{surv}[n]$, and to implement the range target motion compensation expressed in (16).

$$
\begin{aligned}
\text{DFT}_m(\text{IDFT}'_p(S^{CAF}[m, p])') = \text{IDFT}_y(S^*_{ref}[y]) \xi_{surv} \text{DFT}_n\left(\tilde{s}[n], k - \frac{f_{D0} N}{f_s}\right) \\
\times \exp\left(-j2\pi \frac{k}{N} m_{target0} + \phi_{t0}\right) \exp\left(j2\pi \frac{k}{N} \frac{f_{D0} n}{f_c f_s}\right)
\end{aligned}
\quad (14)
$$

$$H_R[k, n] = \exp\left(-j2\pi \frac{k}{N} \frac{f_{D0} n}{f_c f_s}\right) \qquad k = 0, \ldots, N - 1 \quad (15)$$

$$\mathrm{DFT}'_m(\mathrm{IDFT}'_p(S^{CAF}[m,p])') = \mathrm{IDFT}_y(S^*_{ref}[y])\xi_{surv}\mathrm{DFT}_n\left(\tilde{s}[n], k - \frac{f_{D0}N}{f_s}\right)$$
$$\times \exp\left(-j2\pi\frac{k}{N}m_{target0} + \phi_{t0}\right) = \mathrm{IDFT}_y(S^*_{ref}[y]) \cdot \mathrm{DFT}_n(S''_{surv}[n]) \qquad (16)$$

In Figure 9, the scheme to implement the target motion compensation using the CAF generated at each surveillance antenna, the outputs of Doppler filter designed for $f_{Dr}^{c,i}$ closest to the bistatic acceleration in each CPI, and the range migration filter are depicted. The resulted CAFs (17) will present a better integration gain of coherent signal processing despite the considered long integration time.

$$S^{CAF}[m,p]'' = \mathrm{DFT}_p(\mathrm{IDFT}_m(\mathrm{DFT}_m(\mathrm{IDFT}_p(S^{CAF}[m,p])H_D(f_{Dr}^{c,i}))H_R(f_{D0}))$$
$$=\xi^*_{ref} \cdot \xi_{surv} \cdot E_T \cdot \exp\left(\phi_{t0}\right) \qquad (17)$$

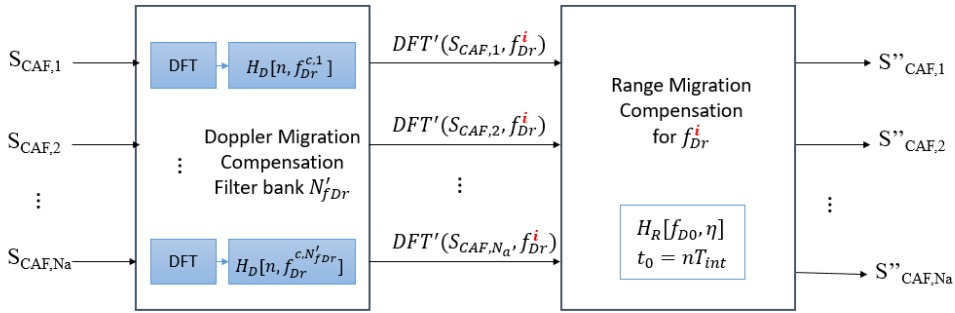

**Figure 9.** Range and Doppler migration compensation scheme.

## 7. Experimental Results

A semi-urban scenario located in the external campus of the University of Alcalá was selected for validating the array architecture. Figure 10 shows the Area of Interest (AoI) for the experiments. The radar scenario was next to Nursing School, with its surveillance area covering a straight road of about 350 m length from the PR, which is surrounded by medium-height buildings, some trees, and a metallic fence that surrounds the sports facilities, where a cooperative vehicle described almost radial movements (Figure 11c).

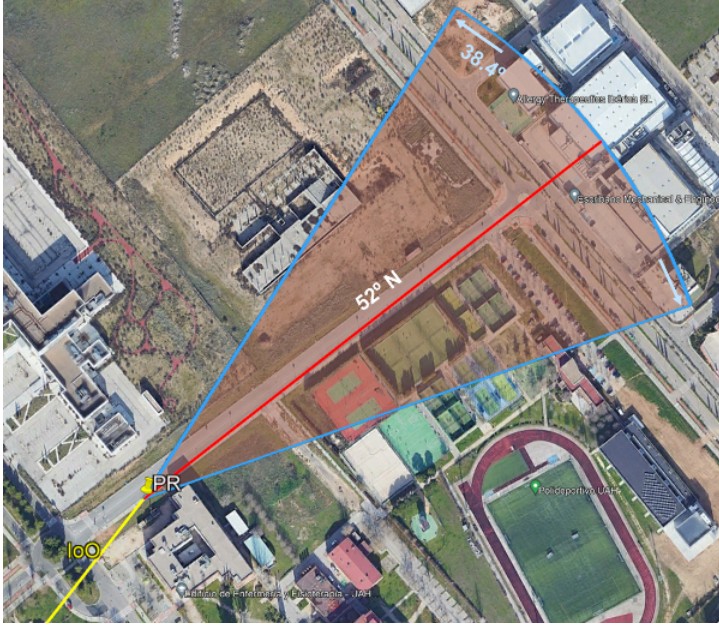

**Figure 10.** Semi-urban scenario for trials located in the external campus of University of Alcaá.

For the experiment, the DVB-S version of IDEPAR was used to acquire real data. In Figure 11a,b, the reference and surveillance antenna systems are depicted: for the reference channel, a commercial parabolic antenna was used with a 40 cm reflector to generate a 5° beamwidth; for the surveillance channel, a 7-element planar array of commercial horn antennas detailed in Section 5.

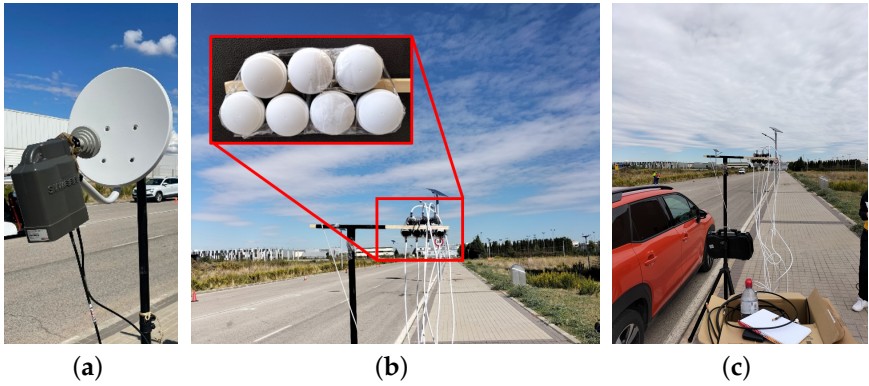

|     (a)     |     (b)     |     (c)     |

**Figure 11.** DVB-S IDEPAR antennas systems and collaborative vehicle in the considered scenario. (**a**) Reference antenna. (**b**) Surveillance antenna array. (**c**) Collaborative vehicle.

The selected IoO is the *Hispasat 30W-5* satellite, which transmits DVB-S signals with an EIRP of 54 dBW. The satellite is seen from the considered scenario at an elevation of 35° and azimuth of 217°N. The scenario geometry shows a quasi mono-static configuration with the PR located between the IoO and the AoI. The selected central frequency is 11.315 GHz, with an acquisition bandwidth of 100 MHz covering 4 horizontally polarized DVB-S channels detailed in Table 1 and presented in Figure 12.

**Table 1.** Adquires DVB-S channels.

| Hispasat 30W-5 (1E) Channels | | | | |
|---|---|---|---|---|
| **Central Frequency** | 11,276 MHz | 11,302 MHz | 11,330 MHz | 11,347 MHz |
| **Polarization** | H | H | H | H |
| **Modulation** | DVB-S2 QPSK | DVB-S2 8PSK | DVB-S2 8PSK | DVB-S2 8PSK |
| **IF** | 1526 MHz | 1552 MHz | 1580 MHz | 1597 MHz |
| **Bandwidth** | 7.140 MHz | 30 MHz | 9.140 MHz | 20.858 MHz |

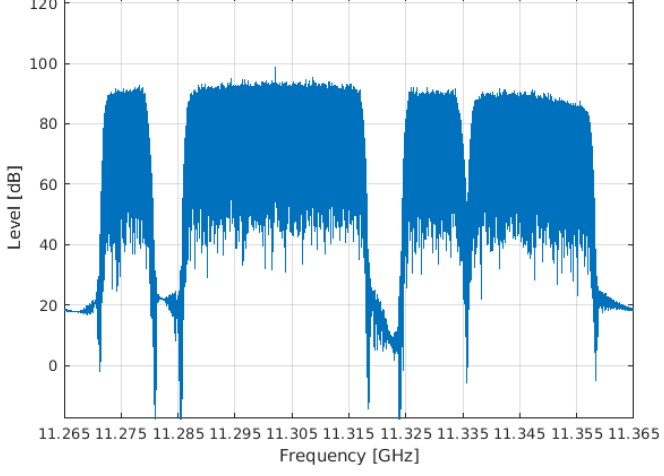

**Figure 12.** Spectrum of acquired data showing the available DVB-S channels.

For each acquisition chain of the surveillance channel, 80 overlapping CPIs were generated using $T_{int}$ = 500 ms and $PRI$ = 250 ms. Doppler Shift and bistatic range resolutions of 2 Hz and 12 m, respectively, were obtained. Surveillance signals were pre-filtered for DVB-S channels selection and noise reduction. To reject DPI and stationary clutter contributions, whose maxima are concentrated along the zero Doppler line of the CAF, the extensive cancellation (ECA) filter was applied. For each CPI and surveillance element, a CAF was generated to apply the proposed processing stages.

Due to waveform characteristics, the ECA filter not only removes the peaks of stationary clutter but also reduces the CAFs pedestal throughout the range-Doppler domain. As a result, DPI contribution will be rejected. In the DVB-S based passive radar, the DPI contribution level is very low for the high propagation losses associated with satellite illuminators, so the implementation of Zero Doppler interference rejection is not a critical stage. For example, Figure 13a,b shows the CAF at a single radiating element level with and without the ECA filter, respectively. The stationary contributions maxima extend along the zero Doppler line up to 200th range bin. The pedestal level reduction is around 4.5 dB.

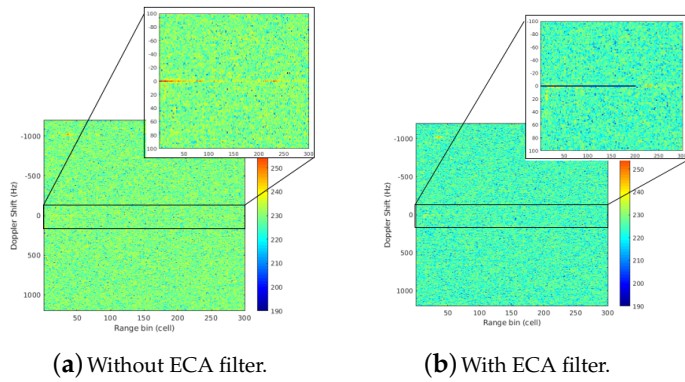

(**a**) Without ECA filter.    (**b**) With ECA filter.

**Figure 13.** Single element CAF of CPI 10.

For the considered scenario, car traffic can be modelled with a mean speed of $v_{tg}$ = 10 m/s ($v_{bis}$ < 20 m/s) and an acceleration of $a_{tg}$ = 1 m/s$^2$, assuming a variation from 0 to 10 m/s in 10 s ($a_{bis}$ < 2 m/s$^2$). Considering the constraints of long integration times described in Section 4, the range and Doppler migration in each CPI are expected because:

- $T_{int}$ > $T_{int,max}$ = 150 ms (for $v_{tg}$ = 10 m/s and $B$ = 100 MHz).
- $a_{tg}$ > $a_{tg,max}$ = 0.053 m/s$^2$ (for $T_{int}$ = 500 ms and $f_c$ = 11.315 GHz).

Following the methodology described in Section 7, the first stage has the objective of target detection and tracking during the acquisition time using the scheme detailed in Figure 6, generating orthogonal beams steering at $\theta = [-10°, -7.2°, -4°, 0°, 4°, -7.2°, -10°]$. The considered $P_{FA}$ for the CFAR detector was set to $10^{-4}$, a high value for improving the detection capabilities of spread target backscattering. A reference and guard 2D-windows with 32 and 4 cells in both dimensions (range and Doppler), respectively, was selected. In Figure 14a,b, the cumulative detection matrix in $T_{acq}$ and the bistatic tracker performance are presented, respectively. The results demonstrate the first approach of the collaborative target trajectory, and the quality parameters can be summarized in $P_D$ = 91.14% ($P_D$ has been estimated at the tracker level, i.e., with the tracker output taking into account the points detected within each track following the methodology detailed in [43]), $P_{FA}$ = $2.65 \cdot 10^{-4}$ ($P_{FA}$ estimation has been done by choosing an area of the detection map where it is assured to be only noise at the pixel level using Monte-Carlo techniques, ensuring an estimation error of less than 10%), $f_{D0}$ = −1018 Hz, and $\overline{f_{Dr}}$ = −12.36 Hz/s (mean value during $T_{acq}$). The declared trajectory presents a non-linear Doppler variation at the beginning of the acquisition (or non-uniform bistatic acceleration variation), where the motion compensation technique could not achieve the best integration gain.

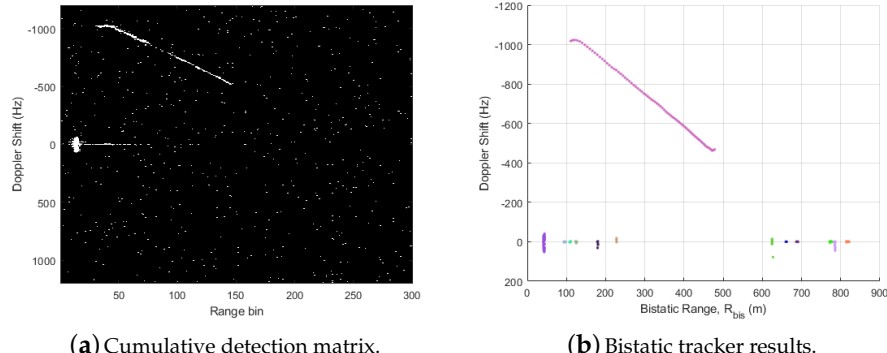

(**a**) Cumulative detection matrix.

(**b**) Bistatic tracker results.

**Figure 14.** Detection and tracking outputs of the first step without target motion compensation of the proposed scheme.

The next step is focused on Doppler migration correction based on a bistatic acceleration filter bank designed for $N_{fDr} = 27$ values from $f_{Drmax} = 94.3$ Hz/s ($a_{bismax} = 2.5$ m/s$^2$) and $f_{Drmin} = -94.37$ Hz/s ($a_{bismin} = -2.5$ m/s$^2$). To improve the understanding of the filter bank bistatic tracking results with $P_{FA} = 10^{-5}$, only performances associated with equispaced $H_D[n, f^i_{Dr}]$ are presented in Figure 15a. The SIR of the collaborative target is high enough to be detected and tracked in all of the $H_D$ filters. Then, the declared trajectories by the filter bank corresponding to the same target are combined to obtain a final result with more accuracy. Focused on the collaborative vehicle, new detection probabilities and target parameters are estimated: $P_D = 98.71\%$, $P_{FA} = 2.81 \cdot 10^{-5}$, $f_{D0} = -1006$ Hz and $\overline{f_{Dr}} = -13.67$ Hz/s. The detection capabilities are clearly improved, and comparison results with respect to the target dynamic estimation in the first stage are depicted in Figure 15b.

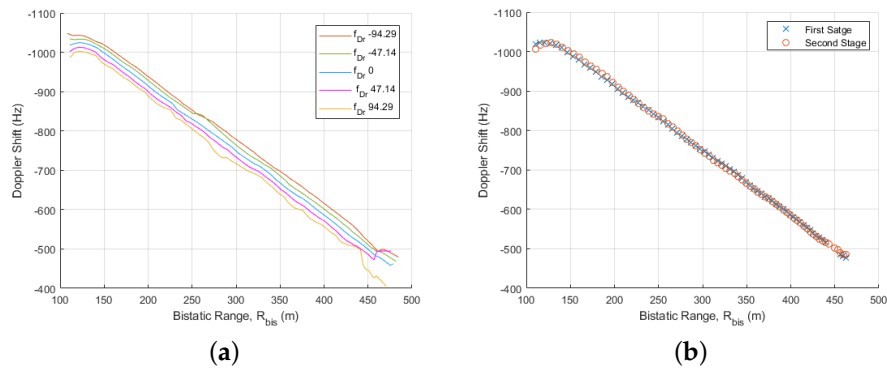

(**a**)

(**b**)

**Figure 15.** Results associated with Doppler migration compensation bank filter. (**a**) Bistatic tracker results for filters designed with $f^i_{Dr} = \{-94.29, -47.14, 0, 47.14, 94.29\}$ Hz/s. (**b**) Tracker performances for the collaborative target are provided by the first and second stages of the proposed scheme.

The new estimated $f_{Dr}$ is used to reduce the filter bank size with $N'_{fDr} < N_{fDr}$, and Doppler walk compensation is performed by selecting, for each CPI, the output of the filter designed for $f^i_{Dr}$ with the highest SIR or with the highest coherent processing integration gain due to the fact $a^i_{bis} = \lambda f^i_{Dr}$ is the closest to the instantaneous target bistatic acceleration $a_{bis}(t = nT_{int})$.

The last stage consists of applying the filter defined in Equation (15) considering the estimated $f_{D0}$ values, for each CPI with $t_0 = nT_{int}$, based on bistatic tracking results presented in Figure 15b. When the output of range walk correction returns to the CAF domain, a new improvement of SIR is expected so the direction of arrival estimation can be implemented with more accuracy. In Figure 16, zoomed areas of the CAFs centered on the collaborative target position on the CPIs= 4, 41, and 72 are presented to show the

SIR improvements in target motion compensation stages and how the target contributions are concentrated on fewer Doppler Shift and range cells. Using the same representation philosophy, zoomed areas of the CAFs centered on the collaborative target position on one CPIs centered at 1250 ms with $T_{int} = 750, 1000$ and 1250 ms are depicted in Figure 17. At the input, the area where target contributions are concentrated is bigger as the integration time is also longer. The Doppler migration reduction can be obtained until CPI value of 1000 ms, but for 1250 ms, a non-linear variation of target Doppler (a non-constant acceleration during CPI) is presented (Figure 17c), and the correction is not so significative. With respect to the range walk, the target trajectory during CPI becomes more complex and non-linear as the time interval increases.

In Figure 18a, the SIR values estimated for the target detections in the CAFs generated for the 0°-steered beam in the first and third stages of the proposed methodology during the 80 CPIs are presented. The results confirm that the target motion compensation provides an SIR improvement of around 10 dB with $T_{int} = 500$ ms. SIR values for CPIs = 4, 41, and 72, and the corresponding DoA accuracy, are summarized in Figure 18b. The SIR after the coherent signal processing is related to the SIR before the integration, the signal bandwidth, and $T_{int}$ [44]. As the input SIR and bandwidth are constant, the SIR improvement with respect $T_{int} = 500$ ms should be 1.76, 3, and 3.98 dB for $T_{int} = 750, 1000$ and 1250 ms, respectively, with the proposed solution when the target dynamics are characterized by constant acceleration and linear movement in RDM domain. The results presented in Figure 19 confirm that controlled long-time intervals can be considered with the proposed methodology to obtain SIR improvements very close to the theoretical ones. The best relationship between CPI value, integration gain, and SIR improvement is obtained for $T_{int} = 500$ ms (Figure 16) and $T_{int} = 750$ ms (Figure 17a).

The final estimated collaborative target trajectory provided by the proposed solution with $T_{int} = 500$ ms is depicted in Figure 20 on 2D map coordinates and compared with GPS data acquired on the collaborative vehicle during the acquisition time, confirming the suitability of the proposed scheme for the DVB-S based on passive radar signal processing.

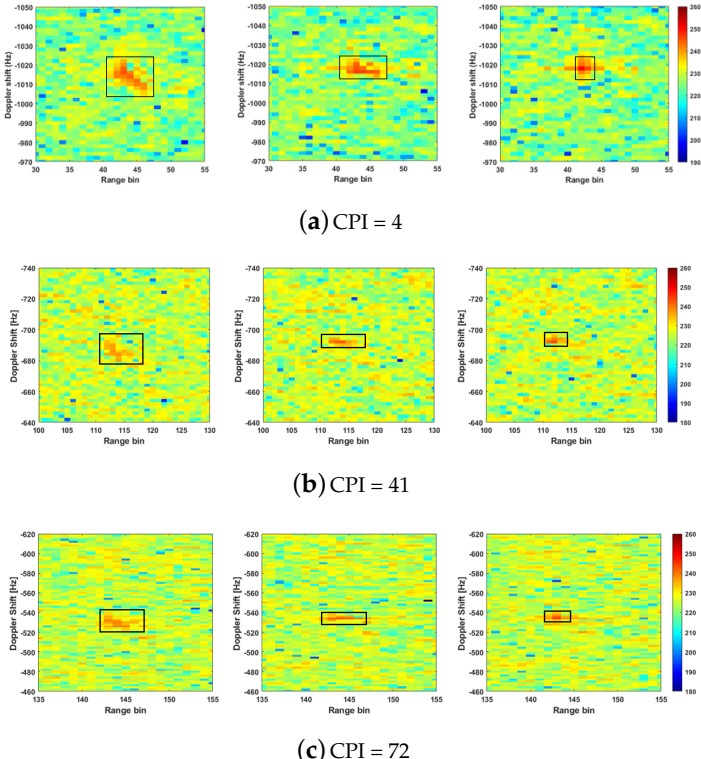

(**a**) CPI = 4

(**b**) CPI = 41

(**c**) CPI = 72

**Figure 16.** Target motion compensation results in the CAF zoomed area in different CPIs. Left, center, and right images are inputs, outputs of second stage, and outputs of third stage of the proposed scheme, respectively.

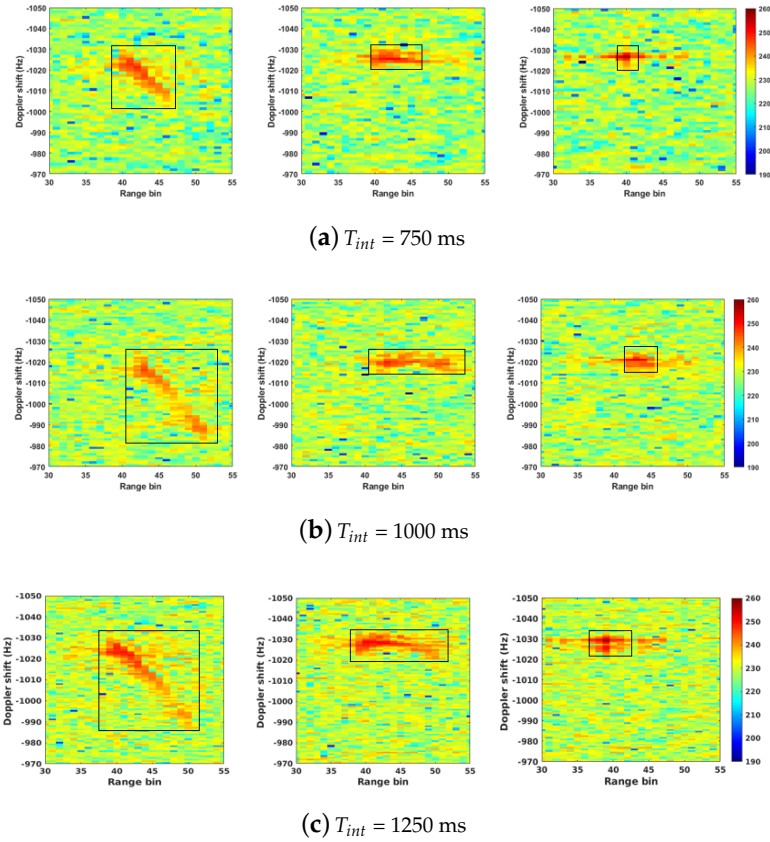

**(a)** $T_{int}$ = 750 ms

**(b)** $T_{int}$ = 1000 ms

**(c)** $T_{int}$ = 1250 ms

**Figure 17.** Target motion compensation results in the CAF zoomed area with different integration times. Left, center, and right images are inputs, outputs of second stage, and outputs of third stage of the proposed scheme.

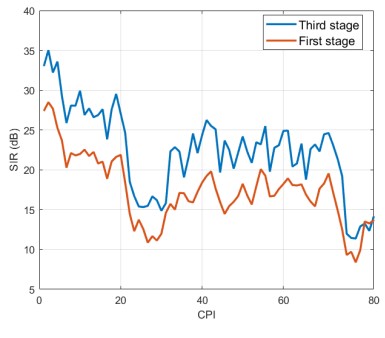

| | | First Stage | Third Stage |
|---|---|---|---|
| CPI = 4 | SIR | 25.26 dB | 33.59 dB |
| | DoA Accuracy | ± 0.0581° | ± 0.0250° |
| CPI = 41 | SIR | 14.47 dB | 23.68 dB |
| | DoA Accuracy | ± 0.2034° | ± 0.0705° |
| CPI = 72 | SIR | 19.52 dB | 24.62 dB |
| | DoA Accuracy | ± 0.1136° | ± 0.0633° |

**(a)** SIR variation during $T_{acq}$        **(b)** SIR and DoA estimation accuracy improvement

**Figure 18.** Study of collaborative target SIR and DoA accuracy measured in CAF generated for the 0°-steered beam associated with the first and third stages of the proposed scheme and $T_{int} = 500$ ms.

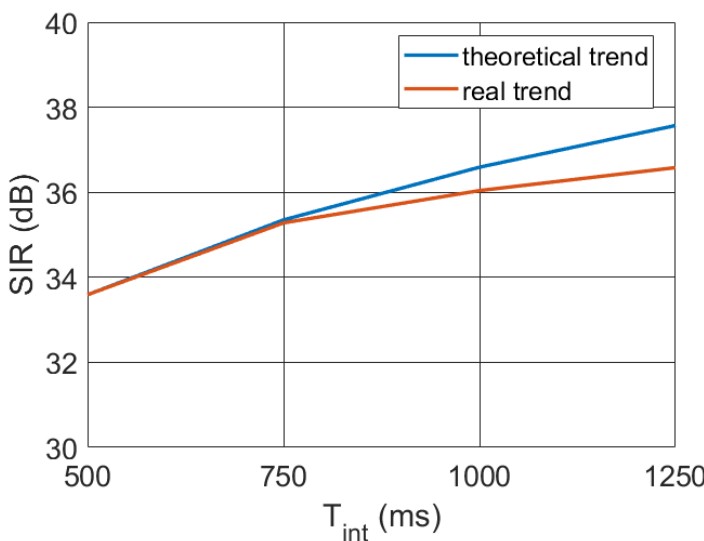

**Figure 19.** Study of SIR improvent versus integration time.

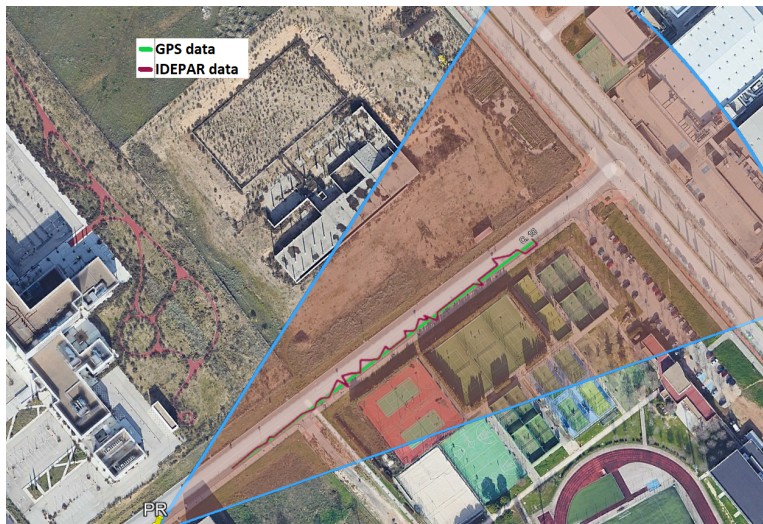

**Figure 20.** Estimated target trajectory with $T_{int} = 500$ ms versus GPS data in geographical coordinates.

## 8. Conclusions

This article tackles the problem of target detection and tracking with a passive radar signal exploiting DVB-S signals. Geostationary satellites IoOs involve high availability but also high propagation losses. The use of the surveillance high-directivity antenna could reduce the azimuth illuminated area, so long integration times for achieving high coherent processing gain are proposed to improve both azimuth and range coverage. However, target dynamics can produce range and Doppler migration (when the target movement during $T_{int}$ exceeds the range or Doppler resolution). To control the corresponding processing gain reduction and to improve DVB-S based PR systems, a multistage target motion compensation technique is proposed.

In the proposed scheme, the detection and tracking techniques are applied at different processing stages to estimate velocities and accelerations and to apply range and Doppler migration compensation adapted to each target. In the first stage without any compensation algorithm, the bistatic tracker results are used to define a set of discrete possible bistatic accelerations. The corresponding Doppler Shift rates are used to design a Doppler migration compensation filter bank that is performed to the IDFT of the CAF generated by each

element of the surveillance antenna array. As SIR improvement is expected, new target trajectories are estimated using the outputs associated with each filter.

Tracks provided by the filters corresponding to the same target are combined using a weighted centroid of target detections and their SIR in each CPI. Then, a final trajectory with more accuracy is obtained for each target to estimate the mean bistatic acceleration during the acquisition time and to reduce the complexity of the Doppler walk correction filter bank. As $f_{Dr}$ can vary during $T_{acq}$, only outputs provided by the filter of the filter bank constrained version with the highest SIR in each CPI are selected. Then, range migration compensation is carried out and SIR is outperformed again. New SIR values lead to an improvement of the detection and tracking capabilities estimating the target direction of arrival with more accuracy.

The proposed scheme is used with data acquired in a semi-urban scenario by IDEPAR, a DVB-S based PR demonstrator developed in the University of Alcalá. As reference and surveillance antenna systems, a commercial parabolic antenna and a 7-element planar array of commercial horn antennas are considered, respectively. The selected IoO signal, provided by *Hispasat 30W-5*, was centered at 11.315 GHz with an acquisition bandwidth of 100 MHz covering 4 DVB-S channels. For the coherent processing signal, a $T_{int} = 500$ ms was considered. The corresponding Doppler Shift and bistatic range resolutions of 2 Hz and 12 m, respectively, were obtained. During the acquisition time, a collaborative target with a GPS device is used for validation purposes.

After the proposed multistage target motion compensation is performed, the results confirm an improvement of SIR and detection capabilities. This trend could lead to an enhancement of the range coverage, which is restricted in satellite PR systems. The target ranges and azimuths estimated in each CPI were depicted in geographical coordinates, and the target trajectory fits very well with the GPS data.

**Author Contributions:** This document has been prepared thanks to the collaboration of the different members of the Acoustic and Electromagnetic Smart Sensor Networks and Signal Processing research team of the University of Alcalá, who have actively collaborated in the study, development, and validation of the proposed methodology. Within this margin, D.M.-M. and M.-P.J.-A. have been in charge of conceptualizing the main milestones of the work carried out. A.A.-H., D.M.-M., N.R.-M. and M.B.-O. have been responsible for the implementation of the multistage solution necessary for the development of the publication. A.A.-H., N.R.-M. and M.B.-O. have been responsible for measurement campaigns and the study of real scenarios. D.M.-M. and M.-P.J.-A. have been responsible for the phase of the project administration, validation, supervision, and funding acquisition. A.A.-H., D.M.-M. and M.-P.J.-A. has been responsible for writing—original draft preparation. A.A.-H., D.M.-M., M.-P.J.-A., N.R.-M. and M.B.-O. have been in charge of writing—review, editing, and visualization. All authors have read and agreed to the published version of the manuscript.

**Funding:** This work has been partially funded by the Spanish Ministry of Science, Innovation, and Universities project RTI2018-101979-B-I00 and PID2021-128898OB-I00, and by the Community of Madrid under project CM/JIN/2021-010.

**Institutional Review Board Statement:** Not applicable.

**Informed Consent Statement:** Not applicable.

**Data Availability Statement:** Not applicable.

**Acknowledgments:** This work has been partially funded by the Spanish Ministry of Science, Innovation, and Universities project RTI2018-101979-B-I00 and PID2021-128898OB-I00, and by the Community of Madrid under project CM/JIN/2021-010.

**Conflicts of Interest:** The funders had no role in the design of the study; in the collection, analyses, or interpretation of the data; in the writing of the manuscript; or in the decision to publish the results.

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
