# Peer review of "Motion Compensation for Long Integration Times and DoA Processing in Passive Radars"

_remotesensing, doi:10.3390/rs15041031_

Round 1
Reviewer 1 Report
1. The authors should be congratulated for writing a very thorough and clear paper, which shows significant improvement in detection probability, and tracking accuracy using real world data.
2. For future research, the authors might like to consider more challenging multiple target scenarios, as would be encountered in the real world for the GEO applications mentioned by the authors, including: vehicles that stop and start and turn and accelerate on roads, as well as multiple closely spaced vehicles, rather than a single car that goes on a straight road at constant speed.
3. For future research, the authors might like to consider the effects of clutter and multipath, both of which would be very challenging in the real world for the GEO scenarios mentioned by the authors.
4. How would the authors deal with the direct signal from the emitters vs. the reflections from the cars of interest, which would be much more challenging for the real world GEO scenarios mentioned by the authors?
5. It would be interesting to see a plot of increase in SIR vs. coherent integration time, for integration times an order of magnitude or two or three greater than studied in this paper.
6. It would be interesting to compare the theoretical optimal improvement in SIR vs. the actual improvement shown in the real world, as a function of integration time.
7. The authors correctly note that the estimation problem is nonlinear owing to the nonlinear coordinate transformation between Cartesian and spherical coordinates. For future research, if the authors attempt longer coherent integration times, there are more advanced algorithms that could be used to mitigate such nonlinearities, including: hybrid coordinates (Raman Mehra, “A comparison of several nonlinear filters for reentry vehicle tracking,” IEEE Trans Automatic Control 1971) and particle flow filters (see the VIDEO https://youtu.be/vqJGB47XoeY, and Liyi Dai & Fred Daum, “On the design of stochastic particle flow filters,” IEEE Transactions on Aerospace and Electronic Systems, October 2022).
Author Response
The authors would like to thank reviewer #1 for the careful and thorough review of the manuscript “Motion compensation for long integration times and DoA processing in passive radars” and for providing us with comments and suggestions to improve the quality of the work. The following document has been prepared to address all comments in a point-by-point fashion.

Reviewer 2 Report
In this work, the authors address the problems arising when long integration times are required. In particular, in the case of the study, a satellite is used as the illuminator of opportunity, and long integration times make sense to compensate for the significant propagation losses. Therefore, during these periods, the target can change the range and/or Doppler cells. The proposed method performs tracking of the target and compensates for these effects in the cross-ambiguity functions. The topic is interesting and the paper is nicely written. However, this reviewer suggests some minor changes to improve the overall quality of the work:
1. In the introduction, a deeper analysis of the advantages and disadvantages of the competitor methods might be useful to the reader. In general, the novel aspects of the work compared to state-of-the-art and their own work [8] should be better motivated.
2. The notation to introduce the speed and the angles could be improved and better motivated by introducing a figure at the beginning of Section 3. Also, I recommend the use of a bold case for vectors, to clarify equations.
3. Some of the acronyms are not defined, thus complicating the readability of the text.
4. The steps leading to eq. (7) could be given in more detail and the differences concerning conventional CAF highlighted.
5. As p is not relevant in equations 9, 10 and 11, it can be removed from the IDFT notation.
6. The tracking results are combined using a weighted centroid. Can the authors provide more details regarding this method?
7. Are the indices for Sref the same in eqs. 12 and 13? What is the meaning of the DFT notation in 14?
8. The validation experiments are conducted in a practical environment, which is very nice. However, it is not clear to this reviewer if some alternatives could be also used to compensate for the movement of the target. If some other methods are feasible, it would be interesting to compare the performance with that of the proposed method.
Author Response
The authors would like to thank reviewer #2 for the careful and thorough review of the manuscript “Motion compensation for long integration times and DoA processing in passive radars” and for providing us with comments and suggestions to improve the quality of the work. The following document has been prepared to address all comments in a point-by-point fashion.

Reviewer 3 Report
In my opinion, this paper is well-written and scientifically sound. I think that it is already qualified for publication in the current form.
Author Response
The authors would like to thank reviewer #3 for the careful and thorough review of the manuscript “Motion compensation for long integration times and DoA processing in passive radars” and for providing us with comments and suggestions to improve the quality of the work. The following document has been prepared to address all comments in a point-by-point fashion.
